# Förster Resonance Energy Transfer Measurements in Living Bacteria for Interaction Studies of BamA with BamD and Inhibitor Identification

**DOI:** 10.3390/cells13221858

**Published:** 2024-11-08

**Authors:** Sebastian Schreiber, Joachim Jose

**Affiliations:** University of Münster, Institute of Pharmaceutical and Medicinal Chemistry, Pharmacampus, 48149 Münster, Germany; sebastian.schreiber@uni-muenster.de

**Keywords:** BAM complex, FRET, flow cytometry, protein–protein interaction, antibacterial, dissociation constant

## Abstract

The β-barrel assembly machinery (BAM) is a multimeric protein complex responsible for the folding of outer membrane proteins in gram-negative bacteria. It is essential for cell survival and outer membrane integrity. Therefore, it is of impact in the context of antibiotic resistance and can serve as a target for the development of new antibiotics. The interaction between two of its subunits, BamA and BamD, is essential for its function. Here, a FRET-based assay to quantify the affinity between these two proteins in living bacterial cells is presented. The method was applied to identify two interaction hotspots at the binding interface. BamD^Y184^ was identified to significantly contribute to the binding between both proteins through hydrophobic interactions and hydrogen bonding. Additionally, two salt bridges formed between BamD^R94^, BamD^R97^, and BamA^E127^ contributed substantially to the binding of BamA to BamD as well. Two peptides (RFIRLN and VAEYYTER) that mimic the amino acid sequence of BamD around the identified hotspots were shown to inhibit the interaction between BamA and BamD in a dose-dependent manner in the upper micromolar range. These two peptides can potentially act as antibiotic enhancers. This shows that the BamA–BamD interaction site can be addressed for the design of protein–protein interaction inhibitors. Additionally, the method, as presented in this study, can be used for further functional studies on interactions within the BAM complex.

## 1. Introduction

Resistance against known antibiotics is progressively spreading. Therefore, new antibacterial entities are urgently needed. According to the WHO, new antibiotics against gram-negative bacteria are in particular need [1]. Common mechanisms of antibiotic resistance in gram-negative bacteria include changes in membrane and cell-wall permeability, biofilm formation, the production of efflux pumps and beta-lactamases, as well as mutation of typical targets [2]. Hence, the identification of new antibacterial entities targeting so far unrealized targets is crucial [3,4]. The β-barrel assembly machinery (BAM) is a multimeric complex responsible for the folding of outer membrane proteins (OMPs) into the outer membrane (OM) of gram-negative bacteria [5]. Several OMPs incorporated by BAM, like OmpA, are necessary for OM integrity, while others, e.g., the autotransporter protein EhaA, are involved in pathogen host interactions [6,7,8]. Consequently, the BAM complex is essential for cell survival and pathogenicity. Additionally, it is located in the OM, and its two essential subunits BamA and BamD are conserved among gram-negative bacteria, altogether making it an attractive target for new antibiotics [3,9]. In contrast to classical targets of antibiotics, which are located in the cytosol, the BAM complex, being located in the OM, makes it easier to access. This is of particular relevance, as a lot of new antibiotic lead compounds with considerable activity in vitro show little or no effect in vivo due to the significant barriers formed by the OM, the bacterial cell wall, and the cytoplasmic membrane [3,4,10]. While the composition of the BAM complex varies among species, in *E. coli* this complex is comprised of five subunits, BamA–BamE [11,12,13]. The central subunit of the complex, BamA, consists of a 16-stranded transmembrane β-barrel and five periplasmatic polypeptide-transport-associated (POTRA) domains. BamA is highly conserved across the family of Enterobacteriaceae, with over 90% sequence identity within the clinically relevant species of *E. coli*, *Klebsiella pneumoniae* and *Enterobacter aerogenes* [14]. The POTRA domains are involved in the initial recognition and binding of nascent OMPs and their delivery to the BAM complex. They also serve as an interaction site for the other subunits of the complex [8,13,15]. Folding of OMPs is catalyzed by the formation of a so-called lateral gate between the β_1_ and β_16_ strand of the BamA β-barrel [16,17,18,19]. While the exact mechanism of OMP folding is the subject of ongoing research, recent structural evidence suggests the formation of a hybrid barrel between the OMP client and the β-barrel of BamA along a lateral gate [16,18,19,20]. The OMP is subsequently released from BamA into the OM. At the same time, the β-barrel of BamA is closed along the lateral gate again. Recently, the cyclic peptide darobactin was identified as a BamA inhibitor, with antimicrobial activity against several clinical isolates [21,22]. Darobactin adopts a β-strand conformation and interacts with BamA by the lateral gate, mimicking the recognition sequence of BamA clients. This is largely mediated by interactions with the backbone of BamA. Darobactin binding to BamA prevents the insertion of BamA client proteins into the outer membrane [21,23]. This underscores the validity of the BAM complex as an antibacterial target. However, due to the possible development of resistances and the high failure rate in the late stages of drug development, further approaches to target the BAM complex need to be explored as well [2,3,4]. In contrast to the catalytically active BamA, the other subunits, BamB, BamC, BamD and BamE, are involved in the coordination and regulation of OMP assembly [24]. These lipoproteins interact with the POTRA domains of BamA. The exact role of the four lipoproteins is still not understood completely; however, only BamD is essential for cell survival [25]. BamD interacts with special recognition sites of incoming OMPs, the so called β-signals, and directs the proteins towards BamA [15,26]. During this process, BamD promotes the early formation of β-strands in the substrate protein and thereby assists in its assembly [15,27]. Interestingly, in vitro, a minimal functional BAM complex consists of BamADE [28]. This is consistent with the fact that BamE stabilizes the interaction between BamA and BamD and supports the productive association of the two proteins [29]. Overall, the inhibition of the interaction between BamA and BamD could be an option to discover new antibiotic entities that are interfering with OMP assembly. In addition to molecules that directly inhibit the activity of BamA, so far only a couple of molecules are known to interfere with the coordination of BamA and BamD. A figure showing BAM inhibitors as described can be found in Appendix A) [30,31]. Among these, the peptide FIRL, as derived from BamD, was shown to interfere with outer membrane integrity. It was hypothesized to do so by inhibiting the interaction between BamA and BamD [32]. Another peptide resembling the C-terminus of BamA bound to BamD and blocked the incorporation of BamA into artificial liposomes in vitro. In the case where it was expressed in the periplasm, it also led to outer membrane permeabilization. However, inhibition of the interaction between BamA and BamD was never experimentally shown [26]. Finally, a small molecule (IMB-H4), identified by high throughput screening using a yeast two hybrid approach, was shown to inhibit the interaction between BamA and BamD [33]. IMB-H4 showed antibacterial activity in vitro and was shown to permeabilize the OM. IMB-H4 binds to BamA, but no specific interaction site is known. All these results support the hypothesis that inhibition of the interaction between BamA and BamD is a viable strategy for the identification of new antibiotics. BamD, however, is interacting with BamA along a wide, relatively flat surface area [12,13]. This is typical for protein–protein interactions (PPIs) and in consequence makes the identification of druggable binding pockets a prerequisite for the rational design of peptides or small molecules that can inhibit the interaction between the two proteins [34,35]. Therefore, we intended to identify such pockets at the BamA – BamD binding interface. For this purpose, we adapted a method developed recently that allowed the quantification of binding between two proteins in living cells [36,37]. Both interaction partners were genetically fused with a fluorescent protein, and the interaction was measured by Förster resonance energy transfer (FRET) after proper normalization of the corresponding fluorescent signals. In this study, we present the development and thorough validation of this strategy for the first time in *E. coli.* It was used for the quantification of the interaction strength between BamA and BamD. We identified two interaction hotspots by introducing alanine mutations into the binding interface of BamD. Subsequently, two peptides, a variant of the FIRL peptide previously known (RFIRLN) and a peptide derived from the amino acid sequence of the second hotspot within BamD (VAEYYTER), were shown to disrupt the binding of BamA to BamD in a dose-dependent manner in living bacterial cells in the micromolar range. These results confirm that the FIRL peptide inhibits the interaction between BamA and BamD, as hypothesized before [32]. In addition, a new druggable binding pocket at the binding interface of BamA and BamD was identified. It can be shown that the interaction between both proteins can be disrupted by a peptide derived from BamD in living cells. These findings could serve as a starting point for the identification of small molecules as inhibitors of the BamA and BamD interaction.

## 2. Materials and Methods

### 2.1. Materials and Reagents

If not stated otherwise, all chemical and components for all media were purchased from Carl Roth GmbH + Co. KG (Karlsruhe, Germany). The purity of reagents is given at their first appearance in the Section 2 Materials and Methods.

### 2.2. Bacterial Strains and Plasmid Construction

*E. coli* Stellar (*F−*, *endA1*, *supE44*, *thi-1*, *recA1*, *relA1*, *gyrA96*, *phoA*, *Φ80d lacZ*Δ *M15*, Δ(*lacZYA-argF*) *U169*, Δ(*mrr-hsdRMS-mcrBC*), Δ*mcrA*, *λ−*) was used for plasmid construction. *E. coli* BL21 (B, *F^−^*, *gal*, *dcm*, *Ion*, *hsdS_B_*(*r_B_^−^ m_B_^−^*), [*malB^+^*]*_K-12_*(*λ^S^*)) was used for expression of all fusion proteins and subsequent flow cytometric analysis. Plasmids were constructed by ligation free cloning using the InFusion™ cloning kit (Clonetech, San Jose, CA, USA) according to the manufacturer’s instructions, based on plasmid backbones described earlier [38,39,40]. Details on the determinants and the construction of each of the used plasmids is given in the Appendix A). Oligonucleotides used in their constructions are listed in Appendix A. Single-point mutations were introduced with the InFusion™ cloning kit. Oligonucleotides used are listed in Appendix A.

### 2.3. Cultivation of E. coli for Flow Cytometry Experiments

At the start of each experiment, *E. coli* BL21 cells were transformed with the corresponding plasmids, spread onto lysogeny broth-miller (LB)-agar plates with 50 µg/mL kanamycin (≥750 I.U./mg) or carbenicillin (purity: ≥88%) and grown over night. It turned out to be crucial to use freshly transformed cells for each experiment. Keeping the cells on agar plates for more than two days or cryoconservation of the cells led to an increase in the variability of the determined parameters K_a_^app.^, *z* and F_max_. The following day, after transformation of the cells, 1 mL of LB medium with 50 µg/mL of the corresponding antibiotic in a 48-well plate was inoculated with a single colony and cultivated overnight (16 h) at 37 °C and 200 rpm. For protein production, M9 mineral medium was used, in which glucose was substituted by 1% (m/m) glycerol (purity: ≥99%) as the carbon source. A quantity of 200 µL of M9 medium in a 96-well plate was inoculated with 2 µL of the overnight culture and cultivated at 28 °C and 200 rpm for 6 h. Then, gene expression was induced with 1 mM L-rhamnose (purity: ≥98%), and afterwards an additional 2 h of cultivation with 0.2% (m/V) arabinose (purity: ≥99%). Cells that carried a plasmid that only contained one promotor were only treated with the corresponding inductor. Following induction of gene expression, cells were cultivated for an additional 16 h at 28 °C and 200 rpm. Cells were subsequently washed with PBS three times and then directly analyzed by flow cytometry.

Peptides (RFIRLN and VAEYYTER) were synthesized by Genicbio with a purity of >95%. For treatment with the peptides, cells were grown as described above. After washing the cells with PBS, they were incubated for 3 h with the appropriate amount of peptide solved in water, as the RFIRLN peptide was not soluble in M9 medium or PBS at concentrations above 1 mM. Cells not treated with the peptide were also incubated in water for 3 h and used as a control for all experiments involving the peptides.

### 2.4. Flow Cytometry

A FACS Aria III flow cytometer (BD Bioscience, Franklin Lages, NJ, USA) with a 70 microns nozzle was used to analyze all cells. A three-channel method was used to measure the donor channel (mNeonGreeen), the acceptor channel (mScarlet-I) and the FRET channel. To measure the mNeonGreen fluorescence, an excitation wavelength of 488 nm and emission wavelength filters of 530/30 nm (band-pass) and 502 nm (long-pass) were used. mScarlet-I fluorescence was excited at 561 nm and measured with emission wavelength filters of 610/20 nm (band-pass) and 600 nm (long-pass). The FRET channel was measured with an excitation wavelength of 488 nm and emission filters of 610/20 nm (band-pass) and 600 nm (long-pass). A maximum of 10,000 cells per second were analyzed. FACSDiva™ Software (BD Bioscience, Franklin Lages, NJ, USA, version 8.0) was used for gating data analysis. For cells only expressing one fluorophore, 50,000 cells per sample were analyzed. For all other samples, a gate (Q2) was applied that selected cells that showed a signal of at least 3000 RFU in both the mNeonGreen and the mScarlet-I channel. A total of 50,000 cells inside the Q2 gate were analyzed. Examples are shown in Appendix A. Cells at the edge of the plot were excluded during calculation of DFRET values and relative donor/acceptor concentrations.

### 2.5. Extraction of Raw Data from Flow Cytometric Experiments

Data points of all cells inside the analysis gate (Q2), as described in *2.4*, were exported as FCS files using FACSDiva™ Software (BD Bioscience, Franklin Lages, NJ, USA, version 8.0). A custom Python program employing the FlowKit [41] was subsequently used to convert the FCS formatted data into tab-delimited txt format.

### 2.6. Calculation of DFRET Data and Fitting of the Binding Model

Calculation of the DFRET and calculation of the relative concentration of the acceptor and donor from the flow cytometric data were performed as described by Hochreiter et al. [37]. Cells with an RFU ≥ 250,000 were excluded from further analysis. Fitting of a binding model based on the law of mass actions to the data was also performed as described by Hochreiter et al. [36,37]. The binding model used here (Equation (1)) fits the affinity (K_a_^app^), the stoichiometry (*z*) and the maximum FRET (F_max_) of a complex. To do so, the relative donor and acceptor concentration, as well as the DFRET, of a large number of cells with different expression levels has to be known. All other equations used for the calculations are described in Appendix A for clarity. The FRET-efficiency of the donor- acceptor tandem, needed for the calculations, was determined by spectral unmixing according to Alexeeva et al. to be 0.43 [42]. All calculations, including the fitting of the binding model using nonlinear regression, were performed with a custom Python program using the *fit* function from the scipy package [43]. In certain parts of the study, one or two of the parameters in the binding model were set to known values, as clearly indicated in the relevant section of the Section 3 Results.
(1)$$DFRET=\frac{-\sqrt{{\left(-don{\;K}_{a}^{app}-\frac{acc}{z}\;K_{a}^{app}-1\right)}^{2}-4\;don\;\frac{acc}{z}{\;K_{a}^{app}}^{2}}+don\;K_{a}^{app}+\frac{acc}{z}\;K_{a}^{app}+1}{2K_{a}^{app}}\times \frac{F_{max}}{don}$$

### 2.7. Depiction of Crystal Structures

All pictures of the BAM complex (Figure 2 and Figure 3) were prepared using MOE (Molecular Operating Environment, Chemical Computing Group, Canada version: 2019.01). A crystal structure of the complete BAM complex was used (PDB: 5D0O); however, only BamA and BamD are shown in Figure 2 and Figure 3. The distances between the attachment points of the fluorescent proteins were measured using the measure tool, implemented in MOE.

## 3. Results

### 3.1. Coupling of BamA with mNeonGreen and BamD with mScarlet-I Enables Reliable Determination of Apparent Affinities by FRET

The affinity between BamA and BamD has not been quantitively measured yet. However, this is a prerequisite to identify amino acids that drive the interaction between both proteins. To quantify the interaction strength between BamA and BamD, a FRET-based interaction assay was established. To develop such an assay, fluorescence labelling of BamA and BamD was necessary. We decided to label BamA with the green fluorescent protein mNeonGreen and BamD with the red fluorescent protein mScarlet-I. This combination of fluorophores has been shown to be an efficient FRET pair before [44]. The fluorescent proteins were genetically fused to the N- or C-terminus of BamA or BamD via a short flexible linker peptide (G_4_S). Accordingly, two fusion proteins of BamA with mNeonGreen and two fusion proteins of BamD with mScarlet-I were generated. This resulted in four different combinations of fusion proteins that could be used for FRET experiments. Four different plasmids were constructed, coding for the possible combinations as mentioned (Appendix A). The different fusion proteins were set under control of two different promotors (an arabinose- and a rhamnose-inducible promotor) on a single plasmid. This way, gene expression of the individual fusion proteins could be precisely controlled. To determine apparent binding affinities in living cells, a binding model (Equation (1)) needs to be fitted to the apparent concentrations of each fluorophore-labeled protein (independent variables) and the calculated amount of energy transfer due to FRET (dependent variable). To derive apparent concentrations from the fluorescence signals and for the calculation of the FRET effect (DFRET), a proper normalization procedure is required to be applied (Appendix A) [37]. Additionally, a sufficient range of apparent concentrations of the donor, as well as of the acceptor, needed to be explored to cover the entire binding curve. A binding model can be fit onto this data by nonlinear regression to determine the apparent affinity constant (K_a_^app.^) for the complex under investigation. Differences in affinity between different complexes, different protein variants (mutations) or the influence of inhibitors on complex stability can subsequently be compared by this procedure. To test if the experimental setup could fulfill the aforementioned requirements, cells expressing combinations of both fusion proteins were analyzed by flow cytometry, as described in the Material and Methods. A broad distribution of cells with different fluorescence intensities was measured for both the mNeonGreen and mScarlet-I fluorescence across all constructs. This resulted in a wide distribution of events in the upper-right quadrant of a 2D contour plot (mNeonGreen vs. mScarlet-I) (Appendix A). Therefore, cells with a wide range of acceptor to donor ratios could be obtained from a single culture. A total of 50,000 events from the upper-right quadrant were analyzed and used for the determination of the K_a_^app.^. To confirm the reproducibility of results obtained by this method, each experiment was performed with five biological replicates (independently transformed and cultured cells) on three different days. The results are shown in Figure 1a. The K_a_^app.^ determined for the C-terminal fusion of mNeonGreen to BamA (BamA-mNeonGreen) and the C-terminal fusion of mScarlet-I to BamD (BamD-mScarlet-I) did not differ significantly in between different days (K_a_^app.^ ≈ 4 × 10^−5^). However, we would like to emphasize at this point that freshly transformed cells were necessary to obtain consistent results. The use of cells stored on agar plates at 4 °C (>2 days) or of cryogenically preserved cells (−80 °C) for the inoculation of the starter cultures resulted in a highly varied K_a_^app.^ determined on different days (Appendix A). The K_a_^app.^ determined with mNeonGreen-BamA and BamD-mScarlet-I did also not differ significantly between different days (K_a_^app.^ ≈ 4 × 10^−5^). Furthermore, no significant differences in K_a_^app.^ in between the measurements performed with the two different FRET pairs were observed. When BamA-mNeonGreen was combined with the N-terminal fusion of mScarlet-I and BamD (mScarlet-I-BamD), the determined K_a_^app.^ value (K_a_^app.^ ≈ 1 × 10^−3^ − 4 × 10^−3^) was significantly higher than the K_a_^app.^ value determined with BamD-mScarlet-I (K_a_^app.^ ≈ 4 × 10^−5^). Furthermore, a significant difference was observed between the K_a_^app.^ values determined on the second day (K_a_^app.^ ≈ 1 × 10^−3^) and the K_a_^app.^ values determined on the first and third day (K_a_^app.^ ≈ 4 × 10^−3^). The interday variability in the results using this construct was therefore considerably higher than with the preceding two constructs. When mNeonGreen-BamA was combined with mScarlet-I-BamD, the K_a_^app.^ value as determined was significantly lower in two of the three experiments (K_a_^app.^ ≈ 10^−9^) than the K_a_^app.^ value determined with the C-terminal fusions of mScarlet-I and BamD (K_a_^app.^ ≈ 4 × 10^−5^). In the third experiment, the determined mean K_a_^app.^ Value (K_a_^app.^ ≈ 4 × 10^−5^) was significantly higher than those from the other two rounds of experiments. Accordingly, the interday variability was again higher with this construct than with the first two FRET constructs. The K_a_^app.^, determined with mScarlet-I-BamD was therefore significantly different when mScarlet-I-BamD was combined with the BamA-mNeonGreen or mNeonGreen-BamA. In contrast, the apparent affinity (K_a_^app.^) determined using BamD-mScarlet-I remained consistent, showing similar K_a_^app.^ values, whether it was paired with mNeonGreen-BamA or BamA-mNeonGreen. The binding model (Equation (1)) used for the determination of K_a_^app.^ also contains a parameter (*z*) to account for the stoichiometry of the complex. During the regression, all non-variables can either be kept at a constant preset value or can iteratively change during the fitting process. We refer to the former as constants and to the latter as parameters. During the regression, *z* can be treated as a parameter or, alternatively, as a constant, that is kept to the known stoichiometry of the complex. Reduction of the parameters that are fit during nonlinear regression is generally preferred, as this increases the stability and reliability of the fitting process. Using fewer parameters makes the model less susceptible to overfitting, which means that the model fits noise in the data rather than fitting the physiochemical phenomena described by the model [45]. However, during assay development, we decided to keep *z* as a parameter rather than a constant in order to check whether solution of the nonlinear regression of the data actually represents the binding event under investigation. If so, nonlinear regression with three parameters (K_a_^app.^, F_max_ and *z*) should result in a value for *z* close to the expected stoichiometry of the complex. In the case of the interaction between BamA und BamD, the stoichiometry is known to be one (1). Hence, the *z* factor was determined alongside the K_a_^app.^ values, as reported in Figure 1a. The results are shown in Figure 1b. The *z* factor describing the stoichiometry for the interaction between BamA-mNeonGreen and BamD-mScarlet-I was determined to be 1.3, 1.6 and 1.2, on average, on three different days, respectively, while the z factor for the interaction between mNeonGreen-BamA and mScarlet I BamD was determined to be, on average, 0.7, 1.1 and 0.7 on three different days. Therefore, in both cases, the determined z factor was relatively close to the expected value of one (1). This is a first indication that the used constructs enable the determination of relative binding affinities between BamA and BamD under the experimental conditions used here. When mNeonGreen-BamA and mScarlet-I-BamD were used for the determination of K_a_^app.^, the *z* factor determined by nonlinear regression was in between −30,000 and 30,000. This is an unrealistic stoichiometry. Presumably, these values only represent a possible mathematical solution of the nonlinear regression. Therefore, it is likely that there was no interaction between mNeonGreen-BamA and mScarlet-I-BamD, and no reasonable binding curve could be fitted to the experimental data. Lastly, the *z* factor determined for the interaction between BamA-mNeonGreen and mScarlet-I-BamD was 1.7, 1.6 and 1.7, on average, in the three independent experiments that were performed. Therefore, the z factor did not match the expected value of one. In addition, the determined K_a_^app.^ values were higher than the values determined with BamD-mScarlet-I. Therefore, it is unclear if the data acquired with the BamA-mNeonGreen and mScarlet-I-BamD FRET pair actually represent a binding event. Taken together, the use of the C-terminal fusion variant of BamD with mScarlet-I appeared to result in more consistent data. We argue that the use of the N-terminal fusion variant (mScarlet-I-BamD) may obstruct the binding between BamA and BamD. This could pose an explanation for why the determined *z* values did not match the expected values. In consequence, we decided to continue the investigation with the BamA-mNeonGreen–BamD-mScarlet-I and mNeonGreen-BamA–BamD-mScarlet-I FRET pairs.

### 3.2. Fluorophore Position and Changes in Affinity Can Be Resolved

Next, we intended to investigate the degree of reliance for the F_max_ parameter. F_max_ is directly correlated with the distance between the fluorophores in the complex (Appendix A). Therefore, we compared the distance between the fluorophores estimated from the crystal structure of the BAM complex (PDB: 5D0O) with the distance estimated from the F_max_ values determined here. In the crystal structure, the distance between the N-terminus of BamA and the C-terminus of BamD was approximately 67 Å, while the distance between the C-terminus of BamA and the C-terminus of BamD was approximately 58 Å. For the calculations as presented, it is important to note that a flexible linker (G_4_S) was introduced in between the fluorophore and the terminus of the respective BAM subunit. This linker allows the fluorescent protein to move relatively freely around the point of attachment. Additionally, the distance from the surface of the fluorescent protein to the fluorophore in the center also contributes to the total distance between both fluorophores. Lastly, the orientation of the fluorophore can influence the maximum FRET efficiency as well. Therefore, it is difficult to determine the exact distance between the fluorophores for both FRET pairs based on Appendix A. Nevertheless, a calculation of relative distances between both FRET pairs can still be quite informative. The mean F_max_ determined over all replicas was significantly different when determined for the mNeonGreen-BamA–BamD-mScarlet-I FRET pair in comparison to the BamA-mNeonGreen–BamD-mScarlet-I pair. The mean F_max_ was 0.28 for the mNeonGreen-BamA–BamD-mScarlet-I FRET pair and 0.36 for BamA-mNeonGreen–BamD-mScarlet-I. The mean distance between the fluorophores of the FRET pairs was subsequently calculated with Appendix A. The result was a distance of 67 Å for the BamA-mNeonGreen–BamD-mScarlet-I FRET pair and a distance of 74 Å for the mNeonGreen-BamA–BamD-mScarlet-I FRET pair. Both distances are larger than the distances derived from the crystal structure, considering the amino acids, which are connected to the linker in each case. However, when one also takes into consideration the flexible linker and the size of the fluorescent proteins, both determined distances are realistic, considering that the fluorophore is approximately 10 Å away from the attachment point of the fluorescent proteins. Additionally, we calculated the difference in the distance between the fluorophores for both FRET pairs for both estimates. This resulted in a difference of 7 Å when using the distances calculated with Appendix A and in a difference of 9 Å when using the measured distances. Hence, the results obtained with both methods are almost identical. Overall, we concluded that in both cases the determined F_max_ values are in good agreement with the measured distances between the attachment points of the fluorophores. In conclusion, when taken together with the determined *z* factor, which was close to the stoichiometry, as expected, a simple overfitting of the model to noise in the data appears to be unlikely. Hence, the experimental setup can be taken to measure apparent affinities between BamA and BamD in living cells.

For a first application of the assay, it was interesting to analyze whether it would be able to measure differences in the affinities of known variants forming the complex. The mutation BamA^E373K^ has been described before to disrupt the binding of BamA to BamD [12,46]. We therefore introduced this mutation into both FRET constructs and determined the K_a_^app.^. All parameters except K_a_^app.^ were fixed at constant pre-set values. The *z* factor was kept at a value of one (1), while the F_max_ was kept at 0.36 for the BamA-mNeonGreen–BamD-mScarlet-I pair and at 0.28 for the mNeonGreen-BamA–BamD-mScarlet-I FRET. When *z* and F_max_ were treated as constants in this experiment, again, no significant difference between the determined K_a_^app.^ for the WT variants of both FRET constructs was overserved (K_a_^app.^ = 1.3 × 10^−4^ and K_a_^app.^ = 1.71 × 10^−4^). When the BamA^E373K^ mutation was introduced into BamA, the determined K_a_^app.^ was significantly lower in both the BamA-mNeonGreen–BamD-mScarlet-I (K_a_^app.^ = 6.3 × 10^−5^) and the mNeonGreen-BamA–BamD-mScarlet-I (K_a_^app.^ = (8.1 × 10^−5^) complex compared to the respective WT variant. Nevertheless, this K_a_^app.^ appeared reasonable, because it was in a similar order of magnitude compared to the K_a_^app^ of the wild type. These results appear to indicate, other than expected, that BamA^E373K^ still interacts with BamD, but this the interaction is weakened. The binding of the BamA^E373K^ variant to BamD as described and analyzed before was accomplished by affinity copurification. Both proteins could not be copurified in case E373 was mutated to K in BamA [46]. This means that, according to the experiments reported earlier, a simple yes or no answer to the question of whether the two proteins interact can be given. If the affinity of a complex drops below a certain threshold, the interaction partners of this complex can no longer be copurified. To set the lower limit of K_a_^app.^ that would be determined by fitting the binding model to experimental data only containing noise, we also measured a FRET construct were mNeonGreen coupled to BamA was combined with free mScarlet-I. FRET signals measured in this construct are only caused by stochastic FRET, and the determined K_a_^app.^ should, therefore, only result from noise in the data. A K_a_^app.^ of 4.4 × 10^−5^ was determined for the mNeonGreen-BamA–mScarlet-I FRET construct. This was significantly lower than both of the K_a_^app.^ determined for the BamA^E373K^ variants. Therefore, our data suggest that BamA^E373K^ is still able to interact with BamD but has a significantly lower affinity to it. These results provide a more quantitative picture on the effect of the E373K mutation on the interaction of BamA and BamD than the copurification experiments. The difference between our control (mNeonGreen-BamA–mScarlet-I) and our interacting samples (mNeonGreen-BamA–BamD-mScarlet-I and BamA-mNeonGreen–BamD-mScarlet-I) with a factor of 3–4 is relatively small. Therefore, the dynamic range of the in vivo affinity determination by this FRET method in bacteria seems to be somewhat limited. Nevertheless, these results also clearly show that differences in the affinity between different variants of BamA or BamD can be resolved with this assay setup.

**Figure 2 cells-13-01858-f002:**
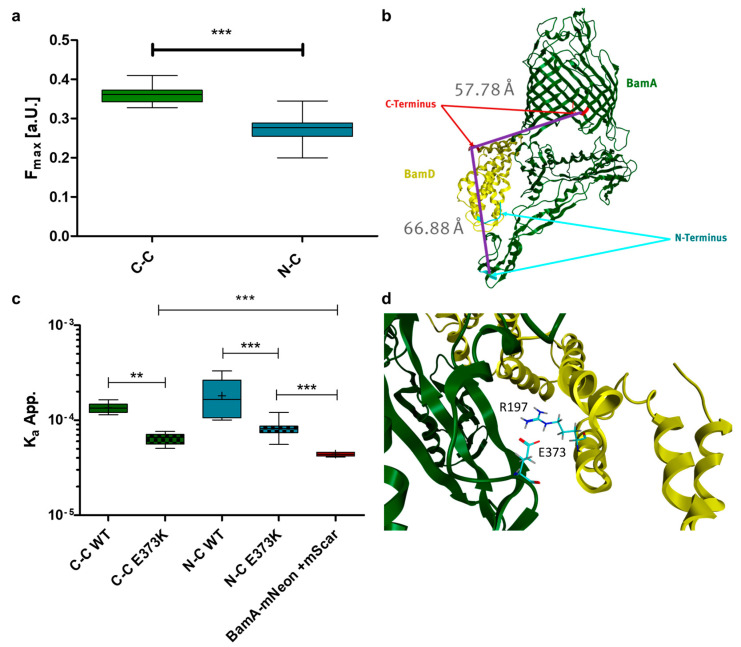
Distance and affinity of the BamA–BamD interaction determined in living cells by fitting a binding model to normalized FRET values. (**a**) The F_max_ value determined by nonlinear regression is shown for the BamA-mNeonGreen–BamD-mScarlet-I construct and the mNeonGreen-BamA–BamD-mScarlet-I construct. Box plots of five biological replicates (n = 5) are shown. Statistical significance was assessed by *t*-test. *** *p* ≤ 0.001 (**b**) Crystal structure (PDB: 5D0O) of BamA (green) and BamD (yellow). The other subunits are not shown. The location of N- and C-termini of both subunits are indicated by blue/red arrows. Fluorophores for the generation of the fusion proteins were fused to either the N- or C-termini via a short flexible linker. The distances between the attachment points of the fluorophores for the BamA-mNeonGreen–BamD-mScarlet-I construct (C-C) and the mNeonGreen-BamA–BamD-mScarlet-I construct (N-C) are also shown. (**c**) Apparent binding affinities (K_a_^app.^) of the BamA-mNeonGreen–BamD-mScarlet-I construct (C-C WT) and the mNeonGreen-BamA–BamD-mScarlet-I construct (N-C WT) and its BamA^E373K^ variants (C-C E373K and N-C E373K), determined by nonlinear regression. The BamA-mNeonGreen–mScarlet-I construct served as a negative control. Box plots of five biological replicates (n = 5) are shown. Statistical significance was assessed by one way ANOVA with Bonferroni pairwise comparison, as indicated in the diagram. ** *p* ≤ 0.01, *** *p* ≤ 0.001 (**d**) Crystal structure (PDB: 5D0O) of BamA (green) and BamD (yellow). The other subunits are not shown. Shown is the salt bridge between BamA^E373^ and BamD^R197^, both colored in cyan.

### 3.3. Identification of Interaction Hotspots in the BamA–BamD Binding Interface

BamA and Bam D interact via two distinct interaction sites. The first is located close to the POTRA 1 domain and the second close to the POTRA 5 domain of BamA (Figure 3a). In the following, the first interaction site—between BamD and POTRA domain 1—will be named site 1, and the second interaction site—between BamD and POTRA domain 5—will be named site 2. The overall surface area of the region comprising site 1 and site 2 is approximately 1470 Å^2^ [47]. Such a large area of contact is typical for protein–protein interactions (PPIs) [48]. Nevertheless, single amino acid contacts can contribute significantly to the PPI and serve as anchor points for the interaction. These so-called “hotspots” could be the target of peptides or small molecules in order to disrupt the PPI [35,49]. Hydrophobic patches can serve as anchor point for PPIs, because desolvation of hydrophobic residues during the binding process is energetically favorable. It has been reported before that Trp, Tyr and Arg contribute disproportionally to PPIs [50]. We identified two neighboring tyrosine residues (Y184, Y185) in an alpha-helical region of BamD that are interacting with site 1 of BamA (Figure 3b). To analyze the contribution of these two amino acids to the overall binding free energy, both were mutated to alanine. The neighboring residues (V181, E183 and E187) that are also interacting with BamA were mutated to alanine as well to elucidate whether these amino acids also contribute to binding to site 1. The K_a_^app^ of all BamD variants for the binding to BamA was determined as described before. The *z* factor and F_max_ values were kept constant during nonlinear regression. Binding of the BamD alanine variants to BamA^E373K^ was determined as a control in every experiment to provide a consistent second reference point, in addition to the binding of BamD to wildtype BamA (first reference point). The K_a_^app.^ values are shown in Figure 3c. No significant difference in affinity was observed when Y185 was mutated to alanine (K_a_^app.^ = 6.22 × 10^−5^) in comparison to the wildtype BamD (K_a_^app.^ = 7.01 × 10^−5^). This is consistent with the fact that Y185 is not involved in any direct interactions with BamA, as seen in the crystal structure. Moreover, the experimental data suggested that neither hydrophobic interactions nor desolvation of Y185 contribute to the overall binding affinity between BamA and BamD in a significant manner. Likewise, none of two further mutations introduced in BamD, E183A (K_a_^app.^ = 7.2 × 10^−5^) and V181A (K_a_^app.^ = 6.66 × 10^−5^), had a significant effect on BamA–BamD binding affinity. Interestingly, mutating E187 to alanine significantly increased the affinity of BamD to BamA (K_a_^app.^ = 7.82 × 10^−5^). The sidechain of E187 is not engaged in any visible hydrogen bond or ionic interaction. The amino acid of BamA closest to E187 is V488. Therefore, BamD^E187A^ might rather increase the hydrophobic contact area between BamA and BamD and could explain the increase in affinity of BamD^E187A^ to BamA. BamD^Y184A^ (K_a_^app.^ = 5.27 × 10^−5^) showed a significantly lower binding affinity to BamA than wildtype BamD (K_a_^app.^ = 7.01 × 10^−5^). Accordingly, from those amino acids analyzed so far, only Y184 of BamD seemed to be an important driving force for the interaction with BamA. In principle, tyrosine can form hydrophobic and π-stacking interactions, as well as hydrogen bonds with its para-hydroxy group. In the crystal structure of the BAM complex, the hydroxy group of BamD^Y184^ is engaged in a hydrogen bond with the backbone carbonyl group of BamA^R336^ (Figure 3c). To determine the share of the possible interactions to the overall binding free energy of binding, Y184 was mutated to phenylalanine. This mutation retains the capability to form hydrophobic and π-stacking interactions but cannot form a hydrogen bond. The relative binding affinity of BamD^Y184F^ (K_a_^app.^ = 6.22 × 10^−5^) was significantly reduced compared to the wild type BamD variant. Accordingly, a hydrogen bond formed by Y184 contributes to the binding of BamD to BamA in a significant manner. However, the relative binding affinity of BamD^Y184F^ was significantly higher than the binding affinity of BamD^Y184A^_._ Hence, in addition to the hydrogen bond, the aromatic ring of Y184 appeared to be engaged in a significant interaction that contributes to the overall binding affinity between BamA and BamD. According to the crystal structure, Y184 does not interact with any aromatic amino acid of BamA. Rather, it seems to interact with the backbone region and the aliphatic part of the sidechain from R366 (Figure 3b), making the contribution of the aromatic ring from Y184 to the overall binding especially noteworthy. Interestingly, it has been reported before that the mutation of BamA^R366^ to a glutamate strongly impaired bacterial growth [12]. R366 is involved in a number of intramolecular ionic interactions. Therefore, it remains obscure whether the growth defect is caused by a disruption of the interaction between BamA and BamD or conformational changes in BamA due to the disruption of the intramolecular interaction network. Nevertheless, it indicates that the interface between BamD^Y185^ and BamA^R366^ could be an interesting interaction hotspot for further investigation.

Mori et al. identified a peptide sequence (FIRL) that was highly conserved across BamD homologues in a wide variety of gram-negative bacteria. The FIRL peptide itself did not show any bacteriostatic or bactericidal effects. However, it potentiated the effect of antibiotics [32]. When the peptide was identified by Mori et al., no crystal structure of the BAM complex was available. When we located the amino acids of the FIRL peptide in the crystal structure of BamD, it turned out that only two of the four amino acids, R97 and L98, appeared to interact with BamA at site 2 (Figure 3c). The sidechains of the other two amino acids, F95 and I96, are directed towards the inner site of BamD, and, therefore, cannot directly interact with BamA. To test if the effects exhibited by the FIRL peptide were at least partially caused by the disruption of the interaction between BamA and BamD, we first analyzed the effect of mutating the corresponding amino acids in BamD. These two amino acids were mutated to quantify their influence on the binding free energy. In addition to that, we extended our analysis to the two neighboring amino acids, one on each side of the FIRL sequence, that were also interaction with BamA. These were R94 and N99. The BamD variants BamD^R94A^, BamD^R97A^, BamD^L98A^ and BamD^N99A^ were generated, and their affinity to BamA was measured as described before. Again, *z* and F_max_ were kept at constant values (*z* = 1, F_max_ = 0.36) during the fitting process of the binding model. The BamA^E373K^ variant was taken as a control in every experiment to provide a consistent second reference point in addition to the wildtype variants (first reference point). The results are shown in Figure 3c. Mutation of R94 to alanine (K_a_^app.^ = 5.47 × 10^−5^) resulted in a significant loss of affinity of the BamD variant to BamA in comparison to the BamD wildtype (K_a_^app.^ = 6.80 × 10^−5^). Judged by the crystal structure, R94 forms a salt bridge with E123 of BamA. The result indicates that the slat bridge formed between R93 and E123 is an important contribution to the overall binding between BamA and BamD. Likewise, BamD^R97A^ (K_a_^app.^ = 6.23 × 10^−5^) also had a significantly lower affinity to BamA than wildtype BamD (K_a_^app.^ = 6.80 × 10^−5^). The effect was less drastic compared to the BamA^E373K^ (K_a_^app.^ = 5.74 × 10^−5^) variant. As seen in the crystal structure, R97 forms a salt bridge with E123 of BamA. Accordingly, the contribution of this salt bridge to the binding free energy between BamA and BamD should also be considerable. BamA^E123^ is involved in a salt bridge with both BamD^R94^ and BamD^R97^. The results presented above indicate that the salt bridge formed between R97 and E123 might be more important than the one formed between R94 and E123. The mutation of L98 to alanine (K_a_^app.^ = 6.58 × 10^−5^) had no significant effect on the affinity of BamD to BamA. This mutation from leucine to alanine does not alter the chemo physical properties of the residue. Therefore, it appears that the alpha methyl group of the alanine, as introduced to BamD, can still form the interactions formed by L98. The BamD^N99A^ variant (K_a_^app.^ = 6.57 × 10^−5^) also showed an affinity to BamA that was not significantly different from wild type BamD. According to the crystal structure, N99 only has a small contact area to A38 of BamA. No hydrogen bonds that are visible in the crystal structure are formed between these two amino acids. In consequence, the results obtained for both variants BamD^L98A^ and BamD^N99A^ appeared plausible.

**Figure 3 cells-13-01858-f003:**
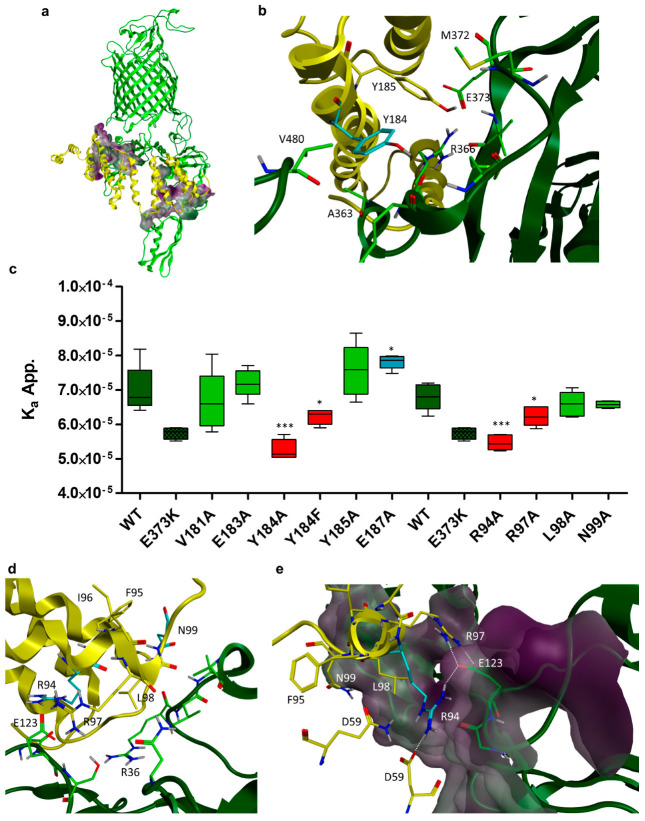
Apparent binding affinities of BamA for different BamD variants determined in living cells by fitting a binding model to normalized FRET values. (**a**) Ribbon model of the crystal structure (PDB: 5D0O) of BamA (green) and BamD (yellow). The other subunits are not shown. Interaction site 1 (upper left) and interaction site 2 (lower right) are indicated by the surface of BamD in proximity to BamA. Green corresponds to lipophilic regions, purple to hydrophilic regions. (**b**) Close up of interaction site 1. Amino acids in the interaction interface that are part of BamA are shown in green. BamD^Y184^ is shown in cyan. BamD^Y185^ is shown in yellow. (**c**) Apparent binding affinities (K_a_^app.^) of the BamA-mNeonGreen–BamD-mScarlet-I construct (WT) and several of its BamD variants, determined by nonlinear regression. The BamA^E373K^ (E373K) variant served as a control. Box plots of five biological replicates (n = 5) are shown. The K_a_^app.^ of variants in which the inside of the box plot is colored in light green do not differ significantly from the wildtype variant. Variants in which the inside of the box plot is colored in red have a K_a_^app.^ that is significantly lower than the wildtype variant. Variants in which the inside of the box plot is colored blue have a K_a_^app.^ that is significantly higher than the wildtype variant. Statistical significance was assessed by one way ANOVA with Dunnett’s test and the WT variant as the control. * *p* ≤ 0.05, *** *p* ≤ 0.001 (**d**) Close up of interaction site 2. Amino acids in the interaction interface that are part of BamA are shown in green. Amino acids of BamD that were part of the FIRL peptide published by Mori et al. are shown in yellow. The neighboring two amino acids that also interact with BamA are shown in cyan. (**e**) Close up of the salt bridge formed between BamD^R97^ (cyan) and BamA^E123^ (green) and BamD^R94^ (yellow) and BamA^E123^ (green). Surface of BamA is shown according to its lipophilicity. Green corresponds to lipophilic regions, purple to hydrophilic regions.

### 3.4. Peptides Derived from BamD Inhibit the Interaction Between BamA and BamD

As described above, we identified two hotspots of the interaction between BamA and BamD, namely BamD^Y184^ and BamD^R94^. Therefore, it was investigated whether peptides derived from the amino acid sequence of the hotspot regions could inhibit the interaction between BamA and BamD. The first peptide that was analyzed, RFIRLN, was similar to the peptide described by Mori et al. [32]. But two extra amino acids were included, an asparagine at the C-terminus of the peptide and an arginine at the N-terminus of the peptide, to mimic the important salt bridge of R94 identified by the alanine scanning. The second peptide, VAEYYTER, was derived from the sequence around Y184. The neighboring three amino acids on either side of the two central tyrosine residues were added to mimic the entire loop region of BamD interacting with BamA (Figure 3a). To test the effect of both peptides on the binding of BamA to BamD, cells expressing BamA-mNeonGreen and BamD-mScarlet-I were treated with different concentrations of the two peptides, and the K_a_^app.^ for the interaction between BamA and BamD was determined as before. Again, the values for *z* and F_max_ were kept constant during nonlinear regression. The results are shown in Figure 4a for RFIRLN and in Figure 4b for VAEYYTER. For both peptides, a clear dose-dependent effect was observed. The K_a_^app.^ determined for cells that were treated with 500 µM of the RFIRLN peptide was significantly lower (K_a_^app.^ = 1.95 × 10^−4^) than for the untreated control (K_a_^app.^ = 3.08 × 10^−4^). Cells expressing mNeon-BamA and mScar-BamD that were treated with 2.5 mM of the peptide showed an even lower K_a_^app.^ of 7.48 × 10^−5^ than cells expressing wildtype BamD and the BamA^E373K^ variant (K_a_^app.^ = 1.73 × 10^−4^). When the cells were treated with 500 µM of the VAEYYTER, the K_a_^app.^ of 1.75 × 10^−4^ was significantly lower than that obtained with the untreated control and in the same range as the K_a_^app.^ of the BamA^E373K^–BamD interaction (K_a_^app.^ = 1.83 × 10^−4^). When the concentration of the VAEYYTER was increased to 2.5 mM, the K_a_^app.^ of the BamA–BamD interaction was 1.03 × 10^−4^_._ Therefore, a dose-dependent effect was also observable for the peptide VAEYYTER. Both peptides showed a similar effect on the interaction between BamA and BamD. However, it is unclear if the peptides competitively inhibit the binding of BamD to BamA, as expected from their sequence. Although both peptides reduced the affinity of BamD to BamA in living cells, no inhibition of bacterial growth was detected up to a concentration of 2.5 mM. The concentrations used here are comparatively high in comparison to concentrations used in other assays, typically in the micro- or nanomolar range. Nevertheless, it is important to note that the bacterial outer membrane is a significant barrier for peptide uptake. Thus, it is unclear how many of the peptides actually entered the cell and reached their target. Cell-penetrating peptides, which contain a number of arginine (and lysine) residues, can be fused to peptide sequences to improve their uptake into cells [51,52]. However, the addition of such cell-penetration residues can influence the binding and the affinity of the original peptide. In this study, our main aim was to identify druggable binding sites in the BamA–BamD interface and show in proof of principle experiment that they can be targeted by drug-like molecules. In consequence, due to the relevance of arginine residues in the binding motif, we abstained from using cell-penetration residues. Nevertheless, the results indicate that addressing the hotspot of the interaction between BamA and BamD is a viable option for the design of PPI inhibitors.

## 4. Discussion

In this study, we developed a FRET-based method for measuring the affinity between BamA and BamD in living cells. We adapted an advanced FRET normalization procedure for the quantitative analysis of PPIs in living cells first described by Hochreiter et al. in eukaryotic cells [36,37,45]. In the original study, the amount of labeled donor and acceptor proteins was adjusted by the amount of DNA used for the transfection of the eucaryotic cells [37]. This required a modification of the method for the present study, as regulation of protein expression by the amount of DNA used for transformation of bacteria is not possible. Instead, we used plasmids with two expression cassettes with different promotors. Extensive validation of the measurements by the variation of fluorophore positions, the estimation of the distances between the fluorophores from measured FRET efficiencies and disruption of the interaction by previously reported mutations supported the validity of the measurements. However, the determined stoichiometry factor of the interaction between BamA and BamD was not one, as theoretically expected. It was further away from the theoretical value of one than reported previously for other protein pairs with a one-to-one stoichiometry [36,37]. Additionally, the dynamic range of the measurement, while sufficient for measuring the influence of mutations on binding affinity, seemed to be somewhat limited. These limitations are likely a result of the relatively high amount of noise in the data. The noise is probably caused by the small size of bacterial cells in comparison to the previously-used eucaryotic cells. The smaller size of the cells reduces the number of proteins present in each cell, and thereby also the absolute amount of FRET that can occur between the fluorescently-labelled proteins when the cells are measured by flow cytometry. Additionally, the smaller size and typically higher expression rates in bacteria compared to eukaryotic cells also increases the concentration of the expressed proteins. This increases the ratio of the signal caused by stochastic FRET and background fluorescence to the signal from FRET specific for the PPI. In turn, this decreases the signal-to-noise ratio. Furthermore, in the present study, both of the proteins are either membrane associated (BamD) or traverse the outer membrane (BamA). It has been reported previously that targeting of noninteracting proteins to membranes results in an increase of spurious FRET [53]. A confined environment like the OM might lead to stochastic FRET, especially at high protein concentrations. This may contribute to the limited dynamic range of the assay, as reported here. Optimization of the FRET pair and/or the expression condition might improve the signal-to-noise ratio and dynamic range of this method in bacteria in the future. Nevertheless, we demonstrated that advanced FRET normalization allows the quantification of the affinity between two proteins in live bacteria. We furthermore showed that the method not only works with soluble proteins but can also be used to investigate integral membrane or membrane-associated proteins.

The interaction between BamA and BamD is essential for the correct function of the BAM complex [28,46,54]. BamE has been reported to stabilize the interaction between BamA and BamD [29]. In this work, BamE was not overexpressed alongside the fluorophore-labeled BamA and BamD fusion proteins. It is therefore unclear to what extent chromosomal-encoded BamE stabilized the complex between fluorophore-labelled BamA and BamD. It is also possible that the interaction measured between BamA and BamD was not affected by BamE at all, due to the low amount of BamE present in comparison to the plasmid-encoded overexpressed fluorophore-labelled BamA and BamD. In a yeast two-hybrid screening conducted by Li et al., BamA and BamD were able to interact in the absence of BamE [33]. Future experiments in a Δ*bamE* strain might help to clarify this. The assay as set up was then used for the identification of interaction hotspots, which could further guide a rational design of PPI inhibitors [35]. Two hotspots were identified in the binding interface between BamA and BamD that are substantially involved in the binding between these two proteins. The first is centered around Y184 of BamD. This amino acid is deeply buried in a hydrophobic cavity of BamA. It is involved in hydrogen bonds and hydrophobic interactions with BamA that significantly contribute to the overall interaction between BamA and BamD. The cavity BamD^Y184^ is binding to is relatively deep, and, therefore, poses a premier opportunity for the design of peptides or small molecules binding at this position. A peptide derived from the sequence of the BamD around Y184 (VAEYYTER) was able to disrupt the binding of BamD to BamA in living cells in high micromolar concentrations. This validated the importance of this region for the PPI and showed the addressability of this pocket by a peptidic inhibitor. However, the potency of the peptide on whole cells needs to improved. Validation of the inhibitory activity and subsequent optimization of the peptide sequence in an in vitro interaction assay could be conducted as a next step. Alongside, the effect of the peptide on OMP assembly by the BAM complex could to be evaluated to ensure that disruption of the interaction between BamA and BamD by the peptide has an effect on the insertase activity of BamA. The second hotspot identified was based on previous results from Mori et al. (2012), who reported a short peptide (FIRL) from BamD to potentiate the activity of antibiotics and cause membrane permeability in treated cells [32]. We showed that amino acids present in the FIRL peptide substantially contributed to the binding between BamA and BamD. At this interaction site, the binding seems to be primarily driven by salt bridges between the two positively-charged residues R94 and R97 of BamD with the negatively-charged residue E127 of BamA. A derived peptide (RFIRLN) was shown to inhibit binding of BamA to BamD in living cells, indicating that the hypothesis of Mori et al. that FIRL could interfere with BamA and BamD binding seems correct. This result also indicates that both interfaces between BamA and BamD can, in principle, be targeted by peptides to disrupt the interaction. Due to the deeper cavity and the hydrophobic nature of the interactions at site one, this region seems to be better suited for the rational design of potential inhibitors. The results, as presented here, lay the ground for the rational design of BamA–BamD interaction inhibitors. Up to now, only a single molecule has been able to disrupt the interaction between BamA and BamD, which was named IMB-H4. It is a small molecule, a 5-nitorfuran derivative, that was identified by high throughput screening with the yeast two-hybrid system. However, no binding site or binding mode was described [33]. The results presented in the present study could serve as possible starting points for the subsequent discovery of new molecules interfering with BamA–BamD interactions.

It is still an open question how successful the strategy of inhibiting the interaction of BamA and BamD could be with regard to the development of new antibiotics. While a lot of data prove the importance of the interaction between these two proteins, there are also mutations known that were generated in lab evolution experiments that enable BamA to function without the need to interact with BamD [15,24,46,55]. It remains open what role these mutations play in an in vivo pathogen host environment and whether these mutations might lead to the rapid development of resistance to BamA–BamD interaction inhibitors in bacteria. Nevertheless, our route for the development of antibacterial compounds presents an alternative approach to target the BAM complex in addition to direct inhibition of the insertase activity of BamA [22,56,57,58]. A number of molecules targeting BamA directly are known. However, only for one of these molecules, a cyclic peptide named darobactin, are a clear mechanism of action, extensive structure activity relationships and an experimentally determined binding mode known [21,23,59,60]. Such information is so far missing for all other molecules targeting the insertase activity of BamA [56,58,61,62]. The binding pockets identified in our study, therefore, provide an additional route for the (semi)-rational design of BAM inhibitors.

In summary, the peptides identified in this study and the methods presented in this paper are useful tools that will not only help in the development of new inhibitors but may also help to further elucidate the complex mechanism of OMP assembly.

## Figures and Tables

**Figure 1 cells-13-01858-f001:**
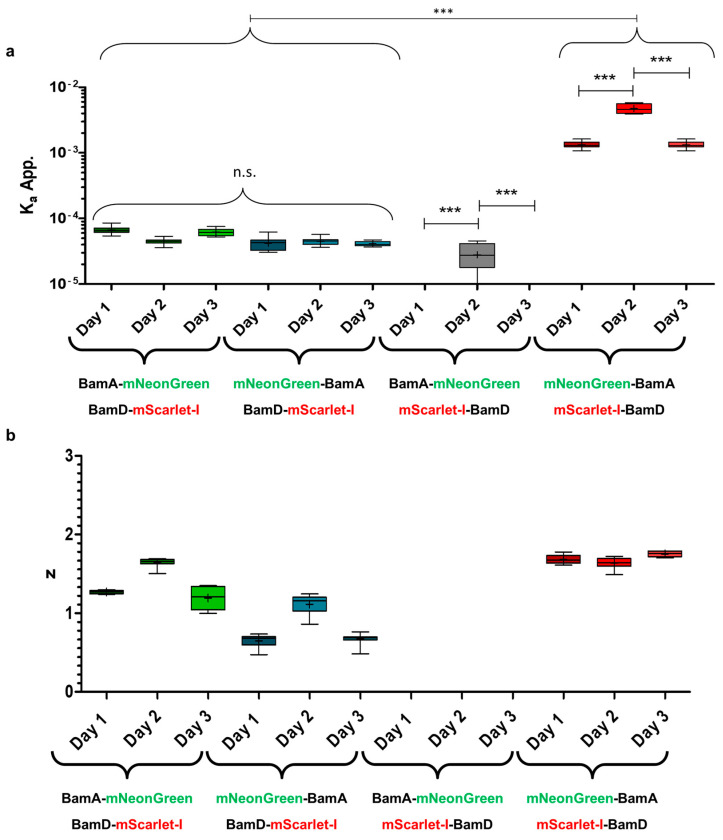
Apparent binding affinities (K_a_^app.^) and stoichiometry factors (*z*) of the interaction between BamA and BamD determined with different FRET pairs on the three different days in living cells by fitting a binding model to normalized FRET values. (**a**) The K_a_^app.^ was determined by nonlinear regression on three different days with five independent samples on each day. Box-plots of five replicates on each day are shown. Four different combinations of the fluorophore position were measured. Statistical significance was assessed by one way ANOVA with Tunkey’s test to compare all pairs of columns. Only significant differences between results from different days within one FRET pair are indicated in the figure (*** *p* ≤ 0.001). The means of all days of the BamA-mNeonGreen–BamD-mScarlet-I and the BamA-mNeonGreen–BamD-mScarlet-I FRET pairs are not significantly different (*p* > 0.05) from each other, as indicate by the brackets. The means of all these samples are significantly different (*** *p* ≤ 0.001) from all samples of the mNeonGreen-BamA–mScarlet-I-BamD FRET pair, as indicate by the brackets. (**b**) The z factor was determined by nonlinear regression on three different days with five independent samples on each day for the same samples as in (**a**). Box-plots of the five replicates on each day are shown.

**Figure 4 cells-13-01858-f004:**
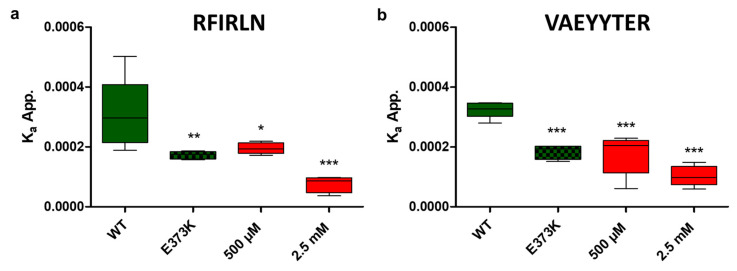
Apparent affinities of BamA for BamD after treatment of the cells with different concentration of the RFILRN (**a**) and VAEYYTER (**b**) peptide, determined in living cells by fitting a binding model to normalized FRET values. BamA^E373K^ (E373K) was measured as a second reference point in addition to the wildtype (WT) variants. Box plots of five biological replicates (n = 5) are shown. Statistical significance was assessed by one way ANOVA with Dunnett’s test and the WT variant as the control. * *p* ≤ 0.05, ** *p* ≤ 0.01, *** *p* ≤ 0.001.

## Data Availability

Raw data are available from the corresponding author upon reasonable request.

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
