# Peer review of "Förster Resonance Energy Transfer Measurements in Living Bacteria for Interaction Studies of BamA with BamD and Inhibitor Identification"

_cells, 2024, doi:10.3390/cells13221858_

Round 1
Reviewer 1 Report
Comments and Suggestions for Authors
Dear Editor the manuscript entitled FRET measurements in living bacteria for interaction studies of 2 BamA with BamD and inhibitor identification has been reviewed now the author needs to address the following points.
Abstract
The author should mention the BamA and BamD interaction significance in the context of antibiotic resistance.
A conclusive line needs to be added regarding the peptides' effectiveness and potential as antibiotic enhancers.
Introduction
Line 25-28:
The author should specify the mechanism of resistance concerned with the Gram-negative bacteria such as efflux pumps, beta-lactamase production, or membrane permeability changes.
The author has written a WHO statement regarding the requirement for new antibiotics against gram-negative bacteria; however reference to the statement is missing author should include a citation.
Line 29-32:
The author should introduce the β-barrel assembly machinery (BAM) as a target for new antibiotics and also discuss why traditional targets become less effective as well as the novelty and potential of BAM
The author should discuss the limitations of existing targets in Gram-negative bacteria
Line 33-35:
The author has mentioned that BAM is highly conserved among gram-negative bacteria kindly provide a brief description
The author should explain the impact of BAM on viability and pathogenicity
Line 36-40:
The author has given the BaMA’s structural features such as 16 stranded transmembrane β-barrel and POTRA domains author should also explain in brief their role in the complex, and how the insertion of proteins into the outer membrane will be done by these structures.
The author should discuss the interaction of darobactin with BamA in detail and also provide a suitable reason for further study on BamA and BamD interaction is important.
Material and methods
FRET-Based Assay Setup (Lines 100-117):
The authors are advised to provide a detailed methodology of FRET-based assay. Mention the excitation and emission wavelength used and the settings of the Flow cytometer as well as the software used for analysis. Also, mention the source and purity grades of key reagents and antibiotics used
Results are presented well
Reviewer 2 Report
Comments and Suggestions for Authors
In the present article two main issues are studied by the authors:
- The interaction between BamA and BamD
- The effect of two peptides which disturb this protein interaction.
The study of BAM is of utmost importance for the future development of new antibiotics.
Abstract
Do you have the IC50 or Ka value for the two peptides? if so, then please add the values in the abstract.
Keywords:
Usually keywords are separated by semicolons, please check the journal requirements.
Introduction =
The introduction is well structured, easy to follow. About 30 references are used to describe the problem and the state of the art.
- The authors mentioned the compound IMB-H4 as a reported inhibitor. But there are other molecules reported too: 10.1093/femsle/fnab059. Would be it possible to incorporate a figure with the chemical structure of the inhibitors reported in literature? I really would appreciate if you can add this information.
Methods:
What software did you use to depict the protein and the aminoacids? Please add the information in the section.
Figure 2d) could you please add the residue code?
Figure 3. could you please add the residue code for b), d), e)?
If the author addressed those issues i think the paper is fully suitable for final publication.
Round 2
Reviewer 1 Report
Comments and Suggestions for Authors
The author has incorporated all the changes can be accepted for publication